# In Vivo Microevolutionary Analysis of a Fatal Case of Rhinofacial and Disseminated Mycosis Due to Azole-Drug-Resistant Candida Species

**DOI:** 10.3390/jof9080815

**Published:** 2023-08-02

**Authors:** Yuchen Wang, Xi Guo, Xinran Zhang, Ping Chen, Wenhui Wang, Shan Hu, Teng Ma, Xingchen Zhou, Dongming Li, Ying Yang

**Affiliations:** 1Bioinformatics Center of AMMS, Beijing Key Laboratory of New Molecular Diagnosis Technologies for Infectious Diseases, Beijing Institute of Microbiology and Epidemiology, Beijing 100850, China; 13716066992@163.com (Y.W.); 13820530560@163.com (X.Z.);; 2TEDA Institute of Biological Sciences and Biotechnology, Nankai University, Tianjing 300457, China; guoxi@nankai.edu.cn; 3Division of Dermatology and Mycological Lab, Peking University Third Hospital, Beijing 100191, China; 4School of Life Science & Technology, China Pharmaceutical University, Nanjing 210009, China

**Keywords:** fungi, microevolution, genetics, candidiasis, rhinofacial and rhino–orbital–cerebral mycosis

## Abstract

Ten *Candida* species strains were isolated from the first known fatal case of rhinofacial and rhino–orbital–cerebral candidiasis. Among them, five strains of *Candida parapsilosis* complex were isolated during the early stage of hospitalization, while five strains of *Candida tropicalis* were isolated in the later stages of the disease. Using whole-genome sequencing, we distinguished the five strains of *C*. *parapsilosis* complex as four *Candida metapsilosis* strains and one *Candida parapsilosis* strain. Antifungal susceptibility testing showed that the five strains of *C. parapsilosis* complex were susceptible to all antifungal drugs, while five *C. tropicalis* strains had high minimum inhibitory concentrations to azoles, whereas antifungal-drug resistance gene analysis revealed the causes of azole resistance in such strains. For the first time, we analyzed the microevolutionary characteristics of pathogenic fungi in human hosts and inferred the infection time and parallel evolution of *C. tropicalis* strains. Molecular clock analysis revealed that azole-resistant *C. tropicalis* infection occurred during the first round of therapy, followed by divergence via parallel evolution in vivo. The presence/absence variations indicated a potential decrease in the virulence of genomes in strains isolated following antifungal drug treatment, despite the absence of observed clinical improvement in the conditions of the patient. These results suggest that genomic analysis could serve as an auxiliary tool in guiding clinical diagnosis and treatment.

## 1. Introduction

Fungal infections affect more than 1 billion people worldwide and cause approximately 1.5 million deaths annually [1]. More than 90% of invasive fungal infections are caused by *Candida* species [2,3]. *Candida albicans* is the most common cause of candidiasis, but infections attributed to other non-*Candida albicans Candida* (NCAC) are of increasing concern. *Candida tropicalis* is one of the most common pathogenic NCAC species in Asia and Latin America [4,5]. *Candida tropicalis* is inherently resistant to azoles [6], one of the main classes of antifungal agents, particularly fluconazole [7,8], and can develop cross-resistance to other antifungal drugs [9]. The molecular mechanisms of azole resistance have been extensively studied in *Candida* species [10,11].

*Candida* species are notorious for causing chronic mucocutaneous candidiasis (CMC) but have never been reported to cause rhinofacial and rhino–orbital–cerebral mycosis (ROCM). CMC is a heterogeneous group of overlapping syndromes characterized by persistent, refractory, and recurrent infections caused by *Candida* species and may include resistant strains [12]. CMC usually occurs when a patient presents with complex deficiencies in immunology, endocrinology, and autoimmunity [13]. This includes a persistent or recurrent infection, and nearly all patients with CMC have inherited or acquired T-cell deficiencies [14]. Immune system deficiencies often render this infection incurable. ROCM is a life-threatening disease that causes facial swelling, ulceration, and disfigurement, with the sudden or progressive onset of symptoms [15]. This infection is associated with high mortality and severe morbidities, such as blindness, eyeball extraction, and facial disfigurement, particularly when the pathogen is drug-resistant or when drug resistance occurs due to the discontinuation of treatment [12,16]. Therefore, caring for patients with ROCM presents significant challenges to physicians because of severe outcomes when the infection is unresponsive to aggressive antifungal therapy.

Whole-genome sequencing (WGS) is gaining increasing importance in the identification and characterization of fungal pathogens [17]. WGS can assist in understanding the genetic diversity and evolution of fungal species, as well as their virulence factors and drug-resistance mechanisms [18]. WGS is currently being implemented in routine diagnostic microbiology, allowing for the identification of antimicrobial resistance genes and genes associated with virulence in bacterial genomes [19], which can be extended to fungal pathogens [20].

Herein, we report a fatal case of CMC with rhinofacial involvement due to the *C. parapsilosis* complex and *C. tropicalis* infection. To further explore the effect of drug therapy on strain evolution, we analyzed the microevolutionary characteristics of multi-drug-resistant *C. tropicalis* in a human host. Analyses of single nucleotide polymorphisms (SNPs), indels, and copy number variants of resistance-associated genes revealed the resistance of the *C. tropicalis* strains to azoles and identified new variants that may lead to antifungal drug resistance. Analysis of mating type locus (MTL) and heterozygous sites demonstrated asexual reproduction and single ancestor and parallel evolution of *C. tropicalis* strains. Molecular clock analysis showed that a change in treatment regimen was consistent with the differentiation time of the strain. We further analyzed presence/absence variations (PAVs) among the *C. tropicalis* strains and found that the strains were evolving toward reduced virulence.

## 2. Materials and Methods

### 2.1. All Candida Strains Culture and Antifungal Susceptibility Assays

All strains used in this study were grown on Sabouraud dextrose agar (SDA) solid medium and cultured in an incubator at 25 °C for 48–72 h. Single colonies were selected and cultured at 37 °C and 200 rpm and activated twice for 16 h each time to ensure that the fungus was in the late exponential growth phase. To investigate in vitro antifungal susceptibility, a broth microdilution method was used according to the M27 [21] and M38 [22] methods of the Clinical and Laboratory Standards Institute (CLSI) for susceptibility testing of nine antifungal drugs (MedChemExpress, Monmouth Junction, NJ, USA). A small number of strains were inoculated into 1 mL of liquid Sabouraud dextrose agar (SDA) medium. The concentration of the strain was determined using a hemocytometer and subsequently diluted to a concentration of 2 × 10^3^ CFU/mL in RPMI 1640 medium (Procell Life Science & Technology Co., Ltd., Wuhan, China). The nine antifungal drugs were gradient diluted and added to 96-well plates. The final concentration gradients are shown in Appendix A. Next, 100 μL of the cell suspension was added to each well at a final concentration of 1 × 10^3^ CFU/mL. After incubation at 37 °C for 24 h, the OD600 of each well was measured using an Epoch Microplate Spectrophotometer (Agilent Technologies, Santa Clara, CA, USA). The CLSI resistance breakpoints of the strains were listed in Appendix A according to CLSI M27 [21] and M59 [23] methods.

### 2.2. Genome Sequencing and Assembly

The methods for library construction, genomic sequencing, and assembly were based on our previous publication [24]. A combined sequencing platform strategy was used for all 10 strains. A 20 kbp single-molecule real-time (SMRT) bell library was prepared from sheared genomic DNA according to the library preparation workflow, and the PacBio Sequel system was used to conduct SMRT sequencing using C3 sequencing chemistry and P5 polymerase with 16 SMRT cells (PacBio, Menlo Park, CA, USA). A Nextera XT DNA Sample Preparation Kit (Illumina Inc., San Diego, CA, USA) was used to prepare the Illumina 250 bp paired-end library with an insert size of approximately 400 bp, and the read inserts were sequenced using Illumina HiSeqXTen (Illumina Inc.). PacBio yielded a total of 22.77 G data with an average coverage of 152× per strain. Illumina yielded a total of 25.56 G data with an average coverage of 170× per strain.

The combination of second- and third-generation sequencing technology can effectively improve the integrity of genome assembly [25]. De novo assembly of PacBio sequencing reads was carried out using the Hierarchical Genome Assembly Process workflow within the SMRT Analysis v2.1 package (www.pacb.com/support/software-downloads, accessed on 26 June 2023). First, error correction of the raw data generated by the PacBio platform was performed using the PacBioRs_PreAssembler with one module, a default minimum subread length of 500 bp, a minimum read quality of 0.80, and a minimum subread length of 7500 bp. Next, the short reads generated by the Illumina HiSeqXTen were mapped to the longer-read PacBio assembly to increase the resolution and quality of the sequences.

### 2.3. Genomic Prediction and Annotation

RepeatModeler [26] and RepeatMasker [27] were used to perform de novo identification and mask repeats. Protein-coding genes were predicted using AUGUSTUS [28], and their functional annotations were assessed according to their homologs in the NCBI non-redundant (nr) database using *E*-value cut-offs of 1 × 10^−5^ and >40% amino acid identity.

### 2.4. Phylogenetic Analysis

Similar to our previous study [29], the phylogeny of the internal transcribed spacer (ITS) region was analyzed using MEGA v7.0.21 [30] with the neighbor-joining (NJ) method with 1000 bootstraps. Orthofinder [31] was used to identify and cluster the orthologous genes. Furthermore, we used Mafft v7.508 [32] to align the multiple sequences. A maximum-likelihood phylogenetic tree was constructed with single-copy orthologs using RAxML v8.2.12 [33], with PROTCATWAG model and 1000 bootstrap replicates. Whole-genome SNPs were used to construct an unrooted neighbor-joining tree with 100 bootstrap replicates using SNPhylo v20180901 [34]. Lastly, ancestral state reconstruction and molecular clock analysis were performed using TreeTime [35] v0.9.0. When the 0/0 and 0/1 genotypes were present in the five *C. tropicalis* strains, the reference allele was identified as the ancestral genotype. When the 1/1 and 0/1 genotypes were present, the alternate allele was identified as the ancestral genotype.

### 2.5. Variant Calling

Low-quality bases from paired-end reads were trimmed using Trimmomatic v0.33 [36], with the following parameters: LEADING:3 TRAILING:3 SLIDINGWINDOW:4:15 MINLEN:80. Paired-end resequencing reads were mapped to the assembly of the 2016_Ct 3 strains with BWA v0.7.10-r789 [37] using the default parameters. SAMtools v1.3.1 [38] was used to convert the mapping results from SAM to the BAM format. Duplicated reads were filtered using the Picard package (Picard. sourceforge. net, v2.1.1) (Broad Institute, 2018, Cambridge, MA, USA). After BWA alignment, the reads around the indels were realigned using the Genome Analysis Toolkit (GATK) v3.8 [39]. Next, the SNP and indel variants for each accession were called using GATK Unified Genotype. Then, the PacBio reads of each sample were mapped to the assembly of the 2016_Ct 3 strains using NGMLR v0.2.7 [40]. The structural variant is called Sniffles. Lastly, functional regions were annotated using the SnpEff v5.1 [41].

### 2.6. Verification of SNPs and Indels

Primer sequences of the resistance genes and DNA mismatch repair genes that were designed using Primer Premier5 (http://www.premierbiosoft.com/primerdesign/, accessed on 26 June 2023) are shown in Appendix A. Sanger sequencing was performed by Beijing Biomed Gene Technology Co., Ltd. (Beijing, China).

### 2.7. Quantification and Statistical Analysis

All details of the statistical analyses are provided alongside the corresponding methods. As described in our previous work [42], PHI database annotation was carried out using blastp, with identity >50% and query coverage >50%. SAMtools v1.3.1 [38] was used to calculate copy numbers. Gene copy number was obtained from gene depth/genome depth. GC skew and gene density were calculated using TBtools v1.098769 [43].

### 2.8. Ethical Declaration

The study was reviewed and approved by Peking University Third Hospital Medical Science Research Ethics Committee IRB (approval: #00006761-2015025).

## 3. Results

### 3.1. Patient Course and Candida Strains

We summarized the disease course of the patient and listed the time points when different antifungal drugs were used and when strains were obtained (Figure 1). *C. tropicalis* strains have been designated as 2015_Ct 1, 2015_Ct 2, 2016_Ct 1, 2016_Ct 2, and 2016_Ct 3. *C. parapsilosis* complex strains have been designated as 2015_Cp, 2015_Cm 1, 2015_Cm 2, 2015_Cm 3, and 2015_Cm 4.

In April 2015, the patient was diagnosed with CMC (ROCM type) based on repeated positive cultures, observation of hyphal forms on direct KOH microscopic examinations, and pathology. Voriconazole was prescribed and resulted in the alleviation and disappearance of symptoms after a 50-day treatment period. However, following the discontinuation of voriconazole and the initiation of corticosteroid treatment, the skin lesions of the patient relapsed. Despite the administration of multiple antifungal drugs, including itraconazole, fluconazole, and voriconazole, the infections worsened, leading to complete damage to the patient’s face. Subsequently, the patient decided to discontinue treatment.

Before the relapse, five *C. parapsilosis* complex strains were isolated, and following the relapse, five *C. tropicalis* strains were isolated. Originally, these strains were characterized as five *C. parapsilosis* and five *C. tropicalis* strains, respectively, using ITS sequencing. Moreover, antifungal susceptibility testing showed that all strains were susceptible to amphotericin B and 5-flucytosine. However, the *C. tropicalis* strains were resistant to the azoles tested, including fluconazole, itraconazole, posaconazole, and voriconazole (Appendix A).

### 3.2. Genome Features and Species Identification

The genomes of all 10 strains were sequenced, and approximately 1.4–5.4 G and 1.3–3.6 G raw data were generated by the PacBio Sequel system and Illumina MiSeq sequencer, respectively. Each genome assembly displayed high continuity, from 11–50 scaffolds with 0.5–2.5 Mb of scaffold N50. The genomic features of all five *C. tropicalis* strains were consistent with those of *C. tropicalis* reference strain MYA-3404 (genome size, <15 Mb; gene number, ~6300; GC content, 33%). However, the genome size of five *C. parapsilosis* strains ranged from 13 Mb (2015_Cp) to 13.9 Mb (2015_Cm 3), and the annotated gene number from 5046 (2015_Cm 2) to 6081 (2015_Cm 3), a difference of approximately 20%. Although strain 2015_Cp possessed the smallest genome, its gene number was intermediate (5548), and it showed the largest gene length (1561 bp), particularly with the GC content (38.69) being slightly higher than that of its partners (Figure 2c).

A maximum-likelihood phylogenetic tree was constructed (Figure 2a) based on 407 single-copy orthologs from 63 *Candida* genomes. All the ITS-characterized *C. tropicalis* strains were clustered with MYA-3404. However, among the ITS-characterized *C. parapsilosis* strains, four belonged to a single clade adjacent to *C. metapsilosis*, whereas 2015_Cp was clustered into another clade comprising *C. parapsilosis* strain CDC317 (Figure 2b). These two clades, along with *C. orthopsilosis*, formed a separate cluster. Previously, *C. parapsilosis* strains were characterized as belonging to three groups (I–III), which were later designated as separate species: Group I remained *C. parapsilosis*, and groups II and III were renamed *C. orthopsilosis* and *C. metapsilosis*, respectively; however, this classification still reflects the close evolutionary relationship among them. Based on this phylogeny, all ITS-characterized *C. parapsilosis* strains were *C. metapsilosis* except 2015_Cp.

### 3.3. Mutations in Drug-Resistance Genes Lead to Azole Resistance

Drug susceptibility testing showed that five *C. tropicalis* strains had high minimum inhibitory concentrations (MICs) to azole antifungal agents, whereas *C. metapsilosis* strains and *C. parapsilosis* strains were susceptible to all antifungal drugs (Figure 3a). We counted the gene mutations (Figure 3b, Appendix A) and copy numbers (Figure 3c) of antifungal resistance genes and their regulators to characterize the relationship between resistance and these genes.

*ERG11* encodes a major component in the membranes of *Candida* species. The *C. parapsilosis* strain 2015_Cp had two single-nucleotide variants (SNVs); however, this mutation did not lead to drug resistance. The two SNVs that lead to amino acid changes (Y132F and S154F), which have been associated with azole resistance in *C. tropicalis*, existed in all our *C. tropicalis* strains. Multiple copies of *ERG11* appeared in all five *C. tropicalis* strains.

*UPC2* is a transcription factor that contributes to fluconazole resistance development by inducing *ERG11* overexpression. In contrast to *C. metapsilosis* strains and *C. parapsilosis* strains containing no mutations in *UPC2,* each of our *C. tropicalis* strains harbored A251T and Q289S amino acid substitutions, except strain 2015_Ct 1, as seen in two fluconazole-resistant strains [44]. Additionally, 2016_Ct 1 and 2016_Ct 2 shared an L168P substitution, which has not previously been recorded. Moreover, strain 2016_Ct 2 harbored a TTAA insertion at position 2334 of the *UPC2* gene, thereby introducing a stop codon and resulting in translation termination at amino acid position 799.

*TAC1* and *MMR1* are two transcription factors that play a role in drug resistance in *Candida* species. Strains 2015_Cm 3 and 2015_Cm 4 had some missense SNPs in *TAC1*, but those did not lead to drug resistance. No SNP mutations were found within *MRR1* in any of the *C. metapsilosis* and *C. parapsilosis* strains. Furthermore, during the screening for *TAC1* (50% protein identity with *TAC1_C. albicans_*) and *MRR1* (57% identity with coding protein of *MRR1_C. albicans_*) in all *C. tropicalis*, we observed 0–4 SNPs resulting in amino acid changes in *TAC1* and 3–11 in *MRR1*, compared with each counterpart of the reference MYA-3404.

We calculated the raw data coverage for each contig to determine whether any large aneuploidy occurred. However, each contig exhibited similar data coverage distribution, indicating that no large aneuploidies occurred (Appendix A).

### 3.4. Strain Differentiation Is Associated with Antifungal Treatment

We analyzed the in vivo microevolution of five multi-drug-resistant *C. tropicalis* strains. K-mer analysis showed that all five strains were diploid (Appendix A), and further heterozygosity analysis revealed that nearly all heterozygous loci (41,070 of 41,892) were shared by all the *C. tropicalis* genomes (Figure 4a), indicating that these strains originated from the same clone.

We identified the MTL to determine the reproduction type of five *C. tropicalis* strains. We found that 2015_Ct 1 and 2016_Ct 2 have the same MTL (**a**/α), whereas 2015_Ct 2, 201605214, and 2016_Ct 3 are a/a mating types. In *C. tropicalis*, **a**/α reproduces asexually; however, α-secreted pheromones can promote homosexual reproduction of **a**/**a** strains to form four ploidy, and such offspring return to diploidy due to instability [45]. All heterozygous loci showed two genotypes, and no allele separation of 0/0, 0/1, and 1/1 occurred, indicating that the strains were propagated asexually (Figure 4b).

Compared with the reconstructed ancestral sequence, the number of “private mutations/total mutations” of the strains, except for 2016_Ct 1, was >50% (Appendix A). Therefore, we identified five *C. tropicalis* strains that underwent parallel evolution in vivo, indicating that *Candida* did not undergo adaptive evolution in the host.

### 3.5. Multi-Drug Resistant Strains Originated from the Same Clone and Underwent Parallel Evolution In Vivo

We determined the haplotype of *C. tropicalis* strains based on the molecular infinite locus model because the strain did not undergo genetic recombination [46]. Based on 822 SNVs among the *C. tropicalis* strains (Appendix A), a maximum-likelihood phylogenetic tree was constructed to determine the ancestral states for each SNV site at the internal nodes (Figure 4c). There are only two-time nodes for calibration; hence, the inferred time will inevitably exhibit some deviation. The putative common ancestor of *C. tropicalis* strains was proposed to have emerged between March 2015 and May 2015 at the start of voriconazole treatment. Within these two months, ancestors quickly diverged into 2015_Ct 2, 2015_Ct 1, and another group. Around September 2015, during corticosteroid treatment, the strains were divided into 2016_Ct 3 and another group. Next, around November 2015, the strains 2016_Ct 1 and 2016_Ct 2 were obtained when the patient started receiving multiple antifungal medications. Through ancestral state reconstruction, we determined that *C. tropicalis* infection developed after the patient was discharged from the hospital rather than having been lurking in the patient until it became the dominant strain.

### 3.6. Abnormal Number of Insertions and Deletions May Be Caused by DNA Mismatch Repair Genes

We analyzed genome-wide variation in the five *C. tropicalis* strains to explore functional changes (Figure 5a). Individual SNV frequency was first investigated, and a high transition rate was observed therein compared to that of transversions. Within transversions, A→T and T→A mutations exhibited a higher frequency, consistent with the comparatively low GC content of the *C. tropicalis* genome (Figure 5d). It is uncommon for more insertions and deletions (indels) than SNVs to be called (Figure 5e). There were 930 indel events, and the number of single-base indels accounted for slightly more than those of multiple indels (Appendix A); 72% of single-base insertions (81/113) and 88% of single-base deletions (335/383) occurred in or near homopolymers.

The deletion of the DNA mismatch repair gene *MSH2* has been confirmed to cause an increase in the occurrence of Indel [47]. The *MSH2* homolog in each of our strains had no mutations except for some synonymous SNVs (Figure 3b). We further investigated other DNA mismatch repair genes and discovered a single-base insertion (2015_Ct 1) or deletion (other *C. tropicalis* strains) in *MUS81* (Appendix A). We found that 2016_Ct 1 had the fewest mutations in DNA mismatch repair genes, and the number of unique indels was significantly lower than that of other *C. tropicalis* strains (Appendix A). Therefore, we speculated that the abnormal number of indels was related to mutations in the DNA mismatch repair genes.

### 3.7. Strains Isolated after Multiple Antifungal Administrations Exhibited Decreased Virulence

Among the multi-drug-resistant *C. tropicalis* strains, the 2015 and 2016 strains were isolated before and after administering multiple antifungals, respectively. We investigated the differences between the strains before and after multiple antifungal administrations.

We characterized the private PAVs of the 2016 strains, defined as the number of structural variations (SVs) that occur in coding sequences and accessible genes (present only in partial strains). In *C. tropicalis* strains, 160 SVs were allocated across 71 genes, including 32 deletions, 100 insertions, and 28 inversions (Figure 5f, Appendix A). The number of accessible genes in 2016_Ct 1 and 2016_Ct 3 was significantly higher than that in the other three *C. tropicalis* strains (Figure 5b). Notably, each *C. tropicalis* strain had a similarly small number of private presence genes (2–4), indicating that the difference in the number of private genes was due to the private absence genes. Among them, 2016_Ct 1 had a maximum of 80 privately absent genes, followed by 2016_Ct 3 with 45 (Figure 5c).

We further annotated strain genomes using the PHI database (Appendix A) to explore changes in the strains after multiple antifungal administrations. We counted PAV events present in the 2016 strains but absent in the 2015 strains. Common PAVs of the three 2016 strains did not show virulence; however, a few depleted genes were strain-specific and closely pathogenicity-related. The four genes depleted in 2016_Ct 1 were *ERG3*, *SOD1*, *BIG1*, and *ADA2*. Uniquely, 2016_Ct 2 deleted two genes, *EAP1* and *CDC42*, which we propose to be virulence-related in *C. tropicalis*. Furthermore, the deletion of *PSY4* appeared only in 2016_Ct 3. Our results indicate that although each strain generated unique absence genes, the strains isolated after multiple antifungal administrations likely generated variants with attenuated pathogenicity under parallel evolution in vivo.

## 4. Discussion

Here, we report the first case of CMC with ROCM involvement due to *Candida* infection. During the early stage of hospitalization, five strains belonging to the *C. parapsilosis* complex were isolated. Using whole-genome sequencing, we successfully differentiated *C. metapsilosis* and *C. parapsilosis*, which could not be distinguished by ITS sequencing. Therefore, we emphasize the importance of utilizing whole-genome sequencing to differentiate the *C. parapsilosis* complex, as it allows for the identification of pathogenic *C. metapsilosis* that arises from mating events between non-pathogenic parental strains [48].

In the later stages of the disease, five azole-drug-resistant strains of *C. tropicalis* were isolated. Although CMC is frequently associated with primary or secondary immunodeficiency, research has shown that drug resistance, innate or acquired, of the strains involved cannot be ignored [12,49]. Multiple copies of *ERG11* appeared in all five *C. tropicalis* strains in this study, which increased the mRNA level, thereby increasing *ERG11* expression. *ERG11* overexpression associated with nonsynonymous point mutations is the most common azole resistance mechanism in *C. tropicalis* [44,50]. Gain-of-function mutations in *UPC2* attributed to certain SNVs have been repeatedly reported to confer azole resistance in *C. albicans* [51,52,53]. The TTAA insertion of strain 2016_Ct 2 is in the extreme C-terminal ligand-binding domain of *UPC2*, the key region for sensing ergosterol levels in the cell [54]. Constitutive upregulation of efflux pumps in azole-resistant *C. albicans* is mediated by gain-of-function point mutations in the zinc cluster transcription factor genes *MRR1* [55,56]. Mutations in *MRR1* may enable *C. tropicalis* to escape an attack from the host, which is likely another role of *MRR1* (*C. albicans*) in increasing *C. albicans* resistance against the innate host defense system by upregulating *MDR1* expression, thereby allowing it to better adapt to certain host niches [57].

We conducted a further microevolutionary analysis of five azole-drug-resistant *C. tropicalis* strains. Sexual reproduction drives the evolution of new traits and adaptation to new environments in eukaryotic organisms, including fungi, allowing for the adoption of different strategies for sexual reproduction [58,59,60]. Analysis of the MTL and heterozygous loci suggests that the strains undergo asexual reproduction, thereby excluding the influence of sexual reproduction on the microevolutionary processes within these strains. We further investigated the parallel evolution of *C. tropicalis* strains and found that the acquisition of azole resistance occurred independently under the drug pressure or was present in the ancestral strains.

Applying molecular clock analysis, we revealed a significant concurrence between the timing of *C. tropicalis* strain differentiation and the change in treatment modality, suggesting a potential impact of drug treatment on the *C. tropicalis* strains. In addition, the ancestral *C. tropicalis* strains emerged around the time of voriconazole treatment, indicating that they were likely drug-resistant and invaded patients during the treatment process, which has recently raised concerns [61]. These *C. tropicalis* strains may have been shed by other patients [62] or have colonized the hospital environment [63]. We thus emphasize the importance of environmental fungal screening and disinfection.

*MUS81* encodes a resolvase for resolving Holliday junctions and has been identified for homologous recombination-mediated DNA repair in *C. albicans* [64]. Indel in *MUS81* results in a frameshift at position 225 and premature termination of translation in all *C. tropicalis* strains, suggesting a loss of *MUS81* function. The absence of DNA mismatch repair genes can enhance genetic diversity and antifungal resistance in fungal strains [65]; however, it can also result in an elevated mortality rate in fungal cells [66].

PAV analysis revealed an absence of virulence genes in *C. tropicalis* strains isolated in 2016 compared to those in *C. tropicalis* strains isolated in 2015. Loss-of-function mutations in *ERG3* increase azole antifungal resistance [67] and attenuate pathogenesis [68] in *C. albicans*. *SOD1*-mutated *C. albicans* is more susceptible to macrophage attachment than the wild type and is less virulent in mice [69]. The *BIG1* gene is involved in beta-1,6 glucan synthesis, and its deletion could reduce synthesis and filamentation, thereby leading to decreased adhesion and virulence in *C. albicans* [70]. *ADA2* is required for drug tolerance and virulence in *C. albicans* and *C. neoformans*, and loss of *ADA2* leads to attenuated virulence. This result contradicts that observed in *C. glabrata*, suggesting the divergent role of *ADA2* in governing pathogenicity between different fungi [71]. *EAP1* helps *C. albicans* to bind to human epithelial cells [72]. The *CDC42*-mutant *C. albicans* cannot form invasive hyphal filaments and germ tubes, resulting in avirulence in a mouse model of systemic infection [73]. Deletion of *PSY4* has been shown to enhance the virulence of *C. albicans* in mice [74]. Overall, our study indicates an evolutionary trend of reduced virulence in the *C. tropicalis* strains.

The absence of DNA mismatch repair genes increases the mortality rate of fungal strains, and the absence of virulence genes leads to microevolution toward reduced virulence. However, clinical observations did not show improvement in patient conditions. This may be attributed to the combination of deep colonization of the strains [75] and limited drug delivery efficiency [10], meaning that longer-term treatment is required.

There were some limitations to this study. Strains representing key clinical time points are scarce in our study, such as the absence of isolates before antifungal treatment, which affects the accuracy of our microevolution analysis. Despite the wealth of research on *C. albicans*, further gene functional experiments are necessary to validate the influence of variations in drug resistance and virulence genes on antifungal drug susceptibility and invasiveness in *C. tropicalis*.

In conclusion, our findings demonstrate a potential evolutionary trend toward reduced virulence in genomes of *C. tropicalis* strains during treatment, despite the absence of clinical improvement in patients’ conditions. Therefore, we suggest that genomic analysis can serve as complementary evidence in guiding clinical treatment decisions.

## Figures and Tables

**Figure 1 jof-09-00815-f001:**
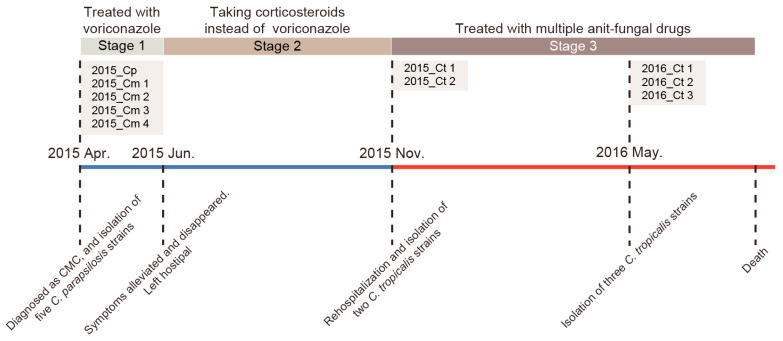
Map of the patient course and strain isolation. The black dotted line represents the turning point in the disease course and the time to obtain the clinical strains. The blue line represents the time before the relapse, and the red line represents the time after the relapse. Isolated strains are labeled above the timeline.

**Figure 2 jof-09-00815-f002:**
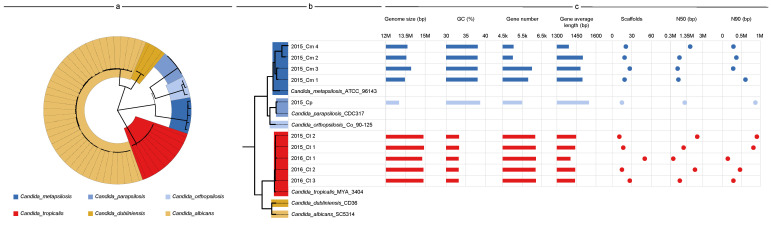
Phylogeny and genome statistics of strains. (**a**) A maximum-likelihood phylogenetic tree of *Candida* strains. Bootstrap was iterated 1000 times. (**b**) Partial representative of the phylogenetic tree. The colors represent different branches, with one reference being selected within each branch. (**c**) General features of genome and sequencing data. Histograms show the genome size, GC content, gene number, and average length. The scatter plots show the number of scaffolds, N50 and N90.

**Figure 3 jof-09-00815-f003:**
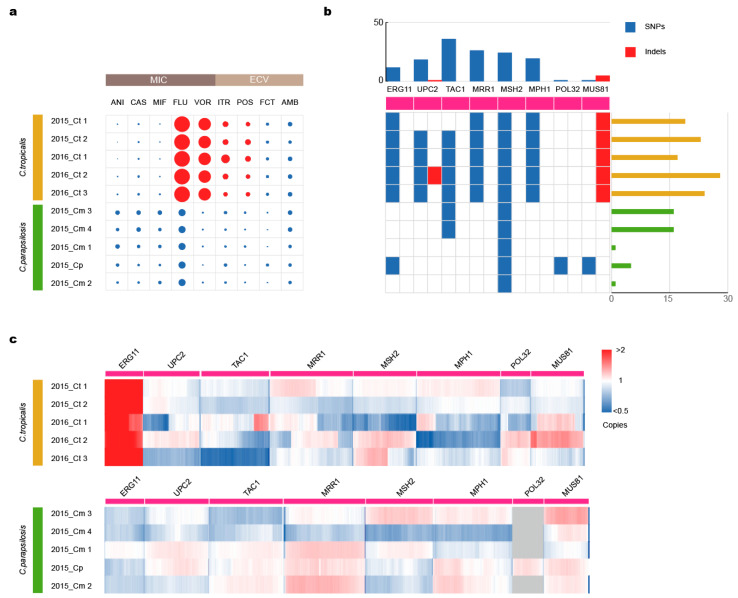
Drug resistance of the strains. (**a**) The MIC or ECV values of nine antifungal drugs. The red circles indicate strains classified as antifungal drug-resistant, and blue circles indicate strains classified as antifungal drug susceptible dose-dependent/intermediate. (**b**) Mutation map of antifungal resistance-associated genes in the strains. The X-axis represents the total number of mutations per strain on eight resistance-related genes, and the Y-axis represents the number of mutations in each resistance-related gene for 10 strains. (**c**) Copy numbers of antifungal resistance-associated genes in the strains. Copy numbers were obtained by comparing the sequencing depth of genomes and single genes using PacBio subreads. The heat map displays upper and lower limits of 2 and 0.5 copies, respectively, corresponding to the changes in the integer copy number.

**Figure 4 jof-09-00815-f004:**
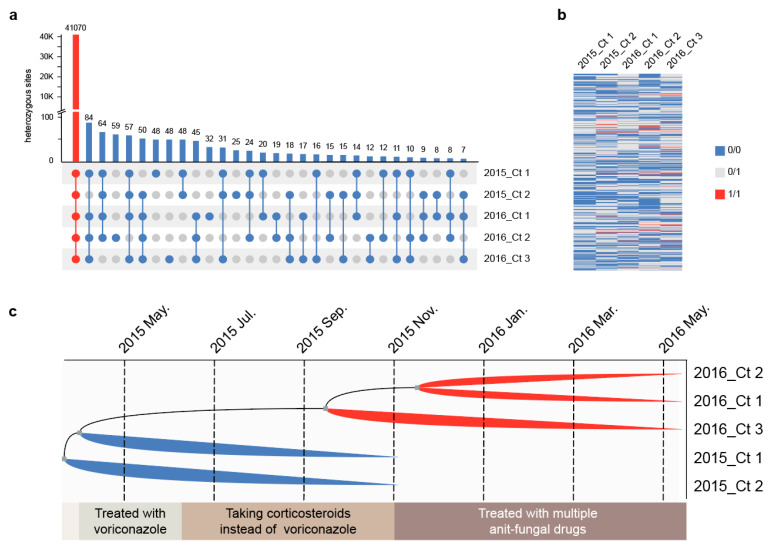
Microevolution of *Candida tropicalis* strains in vivo. (**a**) The number and distribution of common (red) and non-common (blue) heterozygous sites. (**b**) Genotypic distribution of strains at non-common heterozygous loci. A “0/1” represents a heterozygous genotype, and “0/0” or “1/1” means that the individual is homozygous for the reference allele. (**c**) Ancestral state reconstruction of *C. tropicalis* strains. The branch length represents the time scale, gray squares represent divergence nodes, and the bottom box represents the patient course.

**Figure 5 jof-09-00815-f005:**
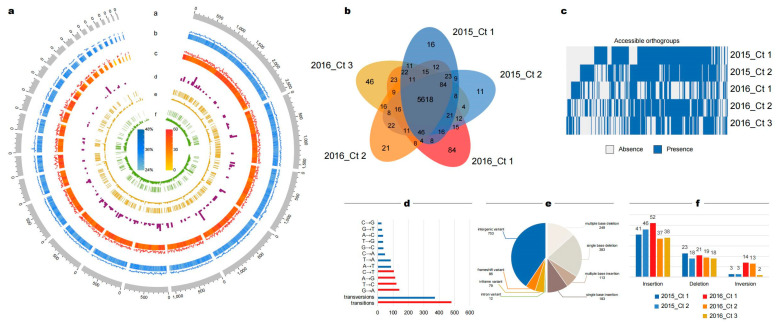
Genome-wide variant events of *Candida tropicalis* strains. (**a**) Distribution of genome features and genetic variations of *C. tropicalis* strains. (a.) Contigs of 2016_Ct 3. (b.) GC content and skew. (c.) Gene density and numbers. (d.) Distribution of SVs. (e.) The yellow vertical bar represents the accessible indel, and the yellow histogram represents the core indel. (f.) The green vertical bar represents the accessible SNVs, and the green histogram represents core SNVs. (**b**) Distribution of orthogroups. Color represents different strains; there are 5618 core orthogroups in the *C. tropicalis* strains. The outermost number represents each strain’s presence/absence of private genes. (**c**) Presence/absence variation among the *C. tropicalis* strains. Blue represents presence, and gray represents absence. (**d**) SNVs distribution. Blue represents transversions, and red transitions. (**e**) Indels distribution. The colors represent different types. (**f**) Differences in structural variation numbers among *C. tropicalis* strains.

## Data Availability

Genome assemblies were deposited into the National Center for Biotechnology Information’s Sequence read Archive under BioProject PRJNA1000740.

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
