# Peer review of "In Vivo Microevolutionary Analysis of a Fatal Case of Rhinofacial and Disseminated Mycosis Due to Azole-Drug-Resistant Candida Species"

_jof, 2023, doi:10.3390/jof9080815_

Round 1

Reviewer 1 Report

The manuscript jof-2499899 “In vivo microevolutionary analysis revealed a fatal case of rhinofacial and disseminated mycosis with multi-drug resistance Candida infection” by Yang et al. reports on a microevolutionary analysis of Candida isolates during a case of chronic mucocutaneous candidiasis (CMC) culminating in death due to rhinofacial involvement.

The authors used a previously adopted (see ref. 19) single cell based sequencing technology to perform a genomic characterization of 10 sequential isolates, 5 Candida parapsilosis species complex and 5 Candida tropicalis, that were obtained from selected skin lesions of the patient between 2015 (when a CMC diagnosis was made) and 2016 (when the patient died).

Based on sequence data, the authors investigated the phylogenetic relationship between the isolates (four of five C. parapsilosis complex isolates were identified as belonging to the C. metapsilosis species) as well as the genetic mechanisms underlying the azole-resistant phenotype of five C. tropicalis isolates. In this context, the authors analyzed single-nucleotide variants to speculate on the microevolution of C. tropicalis isolates during the antifungal treatment courses of the patient.

Unfortunately, despite the potential interest of results here presented, the reviewer noted many problems with the current version of the manuscript that prevent it from being considered for publication. Obviously, the reviewer encourages the authors to modify their manuscript extensively before resubmitting it.

Below are listed some (general or specific) suggestions/comments that the authors need to consider in modifying the manuscript.

1.    Title: Misleading and incorrectly formulated. Multi-drug should be azole drug.

2.    Abstract: Rewrite to clarify/improve sentences (e.g., “From the first fatal case of rhinofacial and rhino-orbital-cerebral mycosis due to Candida infection, 10 Candida spp. were isolated”) as well as to better define how the genome analysis allowed to reach the study’s specific objectives (e.g., “Comparative genomic analysis of the C. tropicalis isolates revealed that the antifungal drug treatment to the patient reported herein was effective.”).

3.    Apart from the Materials and Methods section (which may be considered the sole part of the manuscript that is fairly well written), Introduction should focus on CMC and its clinical manifestations, as well as on the growing importance of WGS analysis in medical mycology. The epidemiology of Candida infections and the antifungal resistance mechanisms are well known to the readers of Journal of Fungi. The last paragraph needs to be rewritten to clarify the aims of the present study, which is the description of a clinical case (albeit relevant). “Our study suggests that genomic analysis should be used to detect the therapeutic efficacy of intractable invasive mycosis and to guide clinical medication and treatment”. Wouldn’t it have been sufficient (and more correct) to perform an in vitro antifungal susceptibility testing (AST) on the Candida isolates at the time of their recovery from skin lesions? “To simulate the drug sensitivity of the isolate in vivo, it was cultured at 37 °C and 200 rpm, and activated twice for 16 h each time, so that the fungus was in the late exponential growth phase. We used a hemocytometer to count and dilute the concentration of the strain”. This seems to be a slightly modified reference AST method rather than a method “to simulate the drug sensitivity of the isolate in vivo”. “Sheared”. Does it mean? How were the 10 isolates stored before analysis?

4.    Results are very difficult to read because they are verbose and contain a lot of information that is, however, relevant to the Materials and Methods or the Discussion.

5.    How was the diagnosis of CMC made? How was ROCM diagnosed? Do “multiple antifungal drugs” mean “multiple treatments with voriconazole”? “The patient discontinued treatment” Why did it occur? Do “10 sets” mean “10 samples from skin lesions”? “Before or after relapse”. Please provide more details on this aspect.

6.    It should not be difficult to simplify the designation of study isolates to clarify the presentation of results. A suggestion would be “2015_isolate 1, 2015_isolate 2, and so on.” and “2016_isolate 1, 2016_isolate 2, and so on”, according to their sequential recovery from the patient’s clinical samples.

7.    “Strong azole resistance” should be “high MICs to azole antifungal agents”.

8.    Discussion is generally poor, and some claims are unsubstantiated.

9.    “The strain used in this study was provided by the Division of Dermatology and Mycological Lab, Peking University Third Hospital, Beijing, China” Does it mean “All isolates used…”?

Author Response

We sincerely thank the editor and all the reviewers for their valuable feedback, which we have used to improve the quality of our manuscript (jof-2499899). The reviewer comments are listed in bold and specific concerns have been numbered. Our responses are in a normal font and changes/additions to the manuscript are in blue.

Question1: Title: Misleading and incorrectly formulated. Multi-drug should be azole drug.

Response: According to your helpful suggestion, we have extensively revised our previous title (line 2-4): “In vivo microevolutionary analysis revealed a fatal case of rhinofacial and disseminated mycosis with azole-drug resistant Candida infection”.

Question2: Abstract: Rewrite to clarify/improve sentences (e.g., “From the first fatal case of rhinofacial and rhino-orbital-cerebral mycosis due to Candida infection, 10 Candida spp. were isolated”) as well as to better define how the genome analysis allowed to reach the study’s specific objectives (e.g., “Comparative genomic analysis of the C. tropicalis isolates revealed that the antifungal drug treatment to the patient reported herein was effective.”).

Response: Thank you for your valuable comments. We have rewritten the abstract to improve the sentence structure and better articulate the significance of our research. The abstract has been modified as follows (line 14-26): “Abstract: Ten Candida species strains were isolated from the first known fatal case of rhinofacial and rhino-orbital-cerebral mycosis caused by Candida infection. Using whole-genome sequencing, we distinguished the Candida parapsilosis complex, which could not be resolved using internal transcribed spacer strain identification. Antifungal susceptibility tests and drug-resistance gene analysis revealed that the five Candida tropicalis strains had high minimum inhibitory concentrations to azole antifungal agents. For the first time, we analyzed the microevolutionary characteristics of pathogenic fungi in human hosts and inferred the infection time and parallel evolution of C. tropicalis strains. Molecular clock analysis revealed that azole resistant Candida tropicalis infection occurred during around the first round of therapy, followed by divergence via parallel evolution in vivo. The presence/absence variations revealed a decrease in virulence of strains isolated following antifungal drug treatment, despite the absence of observed clinical improvement in the conditions of the patient. These results suggest that genomic analysis could serve as an auxiliary tool in guiding clinical diagnosis and treatment.”

Question3: Apart from the Materials and Methods section (which may be considered the sole part of the manuscript that is fairly well written), Introduction should focus on CMC and its clinical manifestations, as well as on the growing importance of WGS analysis in medical mycology. The epidemiology of Candida infections and the antifungal resistance mechanisms are well known to the readers of Journal of Fungi. The last paragraph needs to be rewritten to clarify the aims of the present study, which is the description of a clinical case (albeit relevant). “Our study suggests that genomic analysis should be used to detect the therapeutic efficacy of intractable invasive mycosis and to guide clinical medication and treatment”. Wouldn’t it have been sufficient (and more correct) to perform an in vitro antifungal susceptibility testing (AST) on the Candida isolates at the time of their recovery from skin lesions? “To simulate the drug sensitivity of the isolate in vivo, it was cultured at 37 °C and 200 rpm, and activated twice for 16 h each time, so that the fungus was in the late exponential growth phase. We used a hemocytometer to count and dilute the concentration of the strain”. This seems to be a slightly modified reference AST method rather than a method “to simulate the drug sensitivity of the isolate in vivo”. “Sheared”. Does it mean? How were the 10 isolates stored before analysis?

Response: We are grateful for your professional review work on our article.

In light of your suggestion to focus on chronic mucocutaneous candidiasis (CMC) and its clinical manifestations, as well as the growing importance of whole-genome sequencing (WGS) analysis in medical mycology, we reworked the Introduction. We reduced the content on the epidemiology of Candida infections and antifungal resistance mechanisms, as these topics are likely already well-known to the readers of the Journal of Fungi. We have further elaborated on the clinical cases in the last paragraph to better illustrate the significance of our study.

Our study suggests that genomic analysis should be used to detect the therapeutic efficacy of intractable invasive mycosis and to guide clinical medication and treatment. As you mentioned, treatment based on antifungal susceptibility testing (AST) is the preferred approach. We have thoroughly revised and clarified the descriptions throughout the manuscript. Our objective is to demonstrate that genomic analysis is capable of detecting evolutionary changes, such as the reduction in virulence of isolated strains during the course of treatment. We aim to highlight the significance of genomic analysis as a valuable adjunctive tool in clinical decision-making.

We rewrote “2.1. Strain culture and antifungal susceptibility assays” (line 75) and referenced the M27 [21] and M38 [22] methods of the Clinical and Laboratory Standards Institute (CLSI) to describe our AST method (line 79-83):

“To investigate in vitro antifungal susceptibility, a broth microdilution method was used according to the M27 [21] and M38 [22] methods of the Clinical and Laboratory Standards Institute (CLSI) for susceptibility testing of nine antifungal drugs (MedChemExpress, Monmouth Junction, NJ, USA).

  1. 21. Clinical and Laboratory Standards Institute. Reference Method for Broth Dilution Antifungal Susceptibility Testing of Yeasts, 4th ed. Approved standard M27; Clinical and Laboratory Standards Institute: Wayne, PA, 2017.
  2. 22. Clinical and Laboratory Standards Institute. Reference Method for Broth Dilution Antifungal Susceptibility Testing of Filamentous Fungi; Approved Standard. CLSI Document m38-A2; Clinical and Laboratory Standards Institute: Wayne, PA, 2017.

"Sheared" in the context of "2.2. Genome sequencing and assembly" (line 92) refers to the process of fragmenting genomic DNA into 20 kb libraries during the library preparation workflow of single-molecule real-time (SMRT) sequencing. This fragmentation is achieved by applying shear force through high-speed centrifugation.

Question4: Results are very difficult to read because they are verbose and contain a lot of information that is, however, relevant to the Materials and Methods or the Discussion.

Response: Thank you for your valuable comments. We have appropriately condensed the description of the results section, focusing on the most critical findings. We have also relocated relevant details to the materials and methods or discussion sections, ensuring that each part of the manuscript is organized in a coherent and logical manner.

Question5: How was the diagnosis of CMC made? How was ROCM diagnosed? Do “multiple antifungal drugs” mean “multiple treatments with voriconazole”? “The patient discontinued treatment” Why did it occur? Do “10 sets” mean “10 samples from skin lesions”? “Before or after relapse”. Please provide more details on this aspect.

 Response: We are grateful for your professional review work on our article.

Chronic candidiasis is a disease that involves recurrent or persistent candida infections of the skin, nails, and mucous membranes.

The diagnosis of CMC was confirmed by observation of mycelial forms on direct KOH microscopic examinations, repeated positive cultures and pathology. ROCM was confirmed by the clinical rhino-facial destructive features.

“Multiple antifungal drugs” mean several antifungal drugs, which included itraconazole, fluconazole, as well as voriconazole.

“The patient discontinued treatment” because of his family wanted him to be transferred to another hospital for other treatment.

“10 sets” mean 10 samples from skin lesions .

Question6: It should not be difficult to simplify the designation of study isolates to clarify the presentation of results. A suggestion would be “2015_isolate 1, 2015_isolate 2, and so on.” and “2016_isolate 1, 2016_isolate 2, and so on”, according to their sequential recovery from the patient’s clinical samples.

Response: Thank you for your valuable comment. We have simplified the designation of isolated strains. Isolate 20151105232 has been designated as 2015_Ct 1, isolate 20151105233 as 2015_Ct 2, isolate 20160515214 as 2016_Ct 1, isolate 20160515215 as 2016_Ct 2, and isolate 20160515216 as 2016_Ct 3. Isolate 20150401227 has been designated as 2015_Cm 1, isolate 20150401228 as 2015_Cp, isolate 20150401229 as 2015_Cm 2, isolate 20150403218 as 2015_Cm 3, isolate 20150405224 as 2015_Cm 4.

Question7: “Strong azole resistance” should be “high MICs to azole antifungal agents”.

Response: According to your helpful suggestions, we have revised the “Strong azole resistance” to “high MICs to azole antifungal agents” to clarify the meaning in our article.

Question8: Discussion is generally poor, and some claims are unsubstantiated.

Response: We are grateful for your professional review work on our article. We rewrote the discussion section and cited appropriate references for the claims.

Question9: “The strain used in this study was provided by the Division of Dermatology and Mycological Lab, Peking University Third Hospital, Beijing, China” Does it mean “All isolates used…”?

Response: We are grateful for your professional review work on our article. We have revised the acknowledgements section and highlighted this (line 467-468): “We thank the Division of Dermatology and Mycological Lab, Peking University Third Hospital, Beijing, China for providing all strains used in this study.”

Reviewer 2 Report

This manuscript has been described the possible mechanism of genetics evolution during a fungal infection of one patient, especially in the drug resistant genes. The approach taken by authors is good, using one long case study to describe the genetics evolution of C. tropicalis. However, several points should be considered by authors to improve the manuscript.

Abstract:

Line 20-23: this sentence is difficult to understand. How can we say it is effective treatment, while it can not be observed, and eventually the patient was dead?

Introduction:

Line 69-70: Genomic analysis is not clearly helping the clinical management of patients in the clinical setting, since this approach are far from point-of- care approach which will give more advantage to the patients.

Material and methods:

Line 73: be precise, the fungus is the one that was isolated, not the skin lesion.

Lin 73-87: Please provide reference if there any guideline that was being followed for Strain isolation and antifungal susceptibility assays.

Please mention if the authors obtained the ethical approval from research ethics committee prior from the study.

Results:

I suggest writing only results/data from this study, in the RESULT section. Please not to write/cite other study result in the RESULTS section. Move all discussion, explanation. and interpretation from RESULT section to the DISCUSSION section.

Line 163-164: this is not clear. Please elaborate more how the 10 isolates were isolated. Which strains were before, and which were after relapse. Authors may combine the isolation time of the fungi into the figure 1. This will help readers to understand the evolution according to the time of isolation.

Figure 1: Please give an explanation, how can authors confidently have accepted that the disease course was initiated 7 years prior the 2015 case. Can we justify that the mosquitos’ bites were related with the clinical manifestation and the fungal infection?

Figure 2: Legends in the figure 2 are too small. It cannot be read. Please revise.

figure 3B: Please provide explanation of the axis Y and axis X of the chart. It is very hard to follow.

Discussion:

Please add the discussion of the evolution process itself, to give more explanation to the clinical manifestation. and management of the patient. Some of the explanation has been written in the result section. Author may simply move to the discussion.

Please discuss the association of the antifungal treatment to the evolution that may occur. Should we have considered ineffective or prolonged antifungal intervention may induced the mini evolution? Or just simply because the long course of the disease in this patient give the opportunity to the random mutation, resulting to the evolution?

Conclusion:
Conclusion should be highlight about the evolution of the fungi in the process of treatment. How can treatment result to the Mini evolution?

Round 2

Reviewer 1 Report

The authors Yang et al. provided a revised version of the manuscript jof-2499899 “In vivo microevolutionary analysis revealed a fatal case of rhinofacial and disseminated mycosis with multi-drug resistance Candida infection” along with a cover letter in which they address the reviewers’ comments.

One reviewer found the manuscript much improved. However, some issues remain that need to be addressed by the authors.

o  Title: Please further modify the manuscript as suggested. “In vivo microevolutionary analysis of a fatal case of rhinofacial and disseminated mycosis due to azole-drug resistant Candida species”.

o  Abstract, line 15 “mycosis caused by Candida infection” should be “candidiasis”.

o  Abstract, lines 15–19. Please modify/complete the sentence as “Using whole genome sequencing, we distinguished the five strains initially identified as Candida parapsilosis complex as …”. Please modify the sentence as “Antifungal susceptibility testing showed that the five Candida tropicalis strains had high minimum inhibitory concentrations to azoles, whereas antifungal-drug resistance gene analysis revealed the causes of azole resistance in such strains”.

o  Abstract, line 21. Please modify “Candida tropicalis” as C. tropicalis”.

o  Introduction, line 39. Please use “species” instead of “spp.” throughout the manuscript.

o  Materials and Methods, line 76. Please modify “All strains” as “All Candida strains”.

o  Materials and Methods, line 91. Please add information on the CLSI/EUCAST breakpoints or alternatives (i.e., ECVs/ECOFF values) used to interpret MIC results and to classify the strains as resistant or nonsusceptible to azoles.

o  Results, line 78. Please spell out what “ITS” means if this is first time the authors mention it.

o  Results, line 168. Please modify “mycelial” as “yeast and/or hyphal”.

o  Results, line 180. Please verify if the categorization of all C. tropicalis strains as “resistant” to azoles is according to the MIC breakpoints.

o  Figure 1, Results (paragraph 3.2), and Figure 3. Please spell out what “Cp, Cm, or Ct” mean.

o  Figure 3, lines 251. Please modify as “The red circles indicate strains classified as antifungal drug resistant and blue circles indicate strains classified as antifungal drug susceptible-dose dependent/intermediate” Please check whether “susceptible dose dependent/intermediate” (according to CLSI) should not be “susceptible, increased exposure” (according to EUCAST).
